# Development of a Next-Generation Vaccine Platform for Porcine Epidemic Diarrhea Virus Using a Reverse Genetics System

**DOI:** 10.3390/v14112319

**Published:** 2022-10-22

**Authors:** Guehwan Jang, Duri Lee, Changhee Lee

**Affiliations:** College of Veterinary Medicine, Virus Vaccine Research Center, Gyeongsang National University, Jinju 52828, Korea

**Keywords:** infectious cDNA clone, PEDV, reverse genetics, variant of interest, vaccine platform

## Abstract

For the past three decades, the porcine epidemic diarrhea virus (PEDV) has remained an enormous threat to the South Korean swine industry. The scarcity of an effective method for manipulating viral genomes has impeded research progress in PEDV biology and vaccinology. Here, we report the development of reverse genetics systems using two novel infectious full-length cDNA clones of a Korean highly pathogenic-G2b strain, KNU-141112, and its live attenuated vaccine strain, S DEL5/ORF3, in a bacterial artificial chromosome (BAC) under the control of a eukaryotic promoter. Direct transfection of cells with each recombinant BAC clone induced cytopathic effects and produced infectious progeny. The reconstituted viruses, icKNU-141112 and icS DEL5/ORF3, harboring genetic markers, displayed phenotypic and genotypic properties identical to their respective parental viruses. Using the DNA-launched KNU-141112 infectious cDNA clone as a backbone, two types of recombinant viruses were generated. First, we edited the open reading frame 3 (ORF3) gene, as cell-adapted strains lose full-length ORF3, and replaced this region with an enhanced green fluorescent protein (EGFP) gene to generate icPEDV-EGFP. This mutant virus presented parental virus-like growth kinetics and stably retained robust EGFP expression, indicating that ORF3 is dispensable for PEDV replication in cell culture and is a tolerant location for exogeneous gene acceptance. However, the plaque size and syncytia phenotypes of ORF3-null icPEDV-EGFP were larger than those of icKNU-141112 but similar to ORF3-null icS DEL5/ORF3, suggesting a potential role of ORF3 in PEDV cytopathology. Second, we substituted the spike (S) gene with a heterologous S protein, designated S51, from a variant of interest (VOI), which was the most genetically and phylogenetically distant from KNU-141112. The infectious recombinant VOI, named icPEDV-S51, could be recovered, and the rescued virus showed indistinguishable growth characteristics compared to icKNU-141112. Virus cross-neutralization and structural analyses revealed antigenic differences in S between icKNU-141112 and icPEDV-S51, suggesting that genetic and conformational changes mapped within the neutralizing epitopes of S51 could impair the neutralization capacity and cause considerable immune evasion. Collectively, while the established molecular clones afford convenient, versatile platforms for PEDV genome manipulation, allowing for corroborating the molecular basis of viral replication and pathogenesis, they also provide key infrastructural frameworks for developing new vaccines and coronaviral vectors.

## 1. Introduction

Porcine epidemic diarrhea (PED) lethally affects newborn piglets during their first week of life, causing severe watery diarrhea, fatal dehydration, and high mortality [1,2]. PED virus (PEDV), an etiological agent responsible for PED, is a swine coronavirus belonging to the subgenus *Pedacovirus* of the genus Alphacoronavirus within the family *Coronaviridae* of the order *Nidovirales* [3]. The virus has a single-stranded, positive-sense RNA genome of about ~28 kb comprising at least seven open reading frames (ORFs). The first large ORF1a and ORF1b encode the polyproteins 1a and lab, which proteolytically mature to produce 16 nonstructural proteins (nsp1–16). Conversely, the remaining ORFs encode four canonical coronaviral structural proteins, the spike (S), envelope (E), membrane (M), and nucleocapsid (N) proteins, and a single accessory gene, ORF3 [1,4,5]. Based on S gene phylogeny, PEDV strains can be grouped into two main genotypes with two sub-genotypes: low pathogenic (LP) genotype 1 (classical G1a and recombinant G1b) and highly pathogenic (HP) genotype 2 (local epidemic G2a and global epidemic or pandemic G2b) [1,2].

Since the 2013–2014 PEDV pandemic originating in North America, this swine enteric coronavirus has posed a major health and economic threat to the global pig population [1]. Thus, a large amount of research effort has been invested into controlling PED in the field, and accordingly, substantial information about the molecular epidemiology and genetic evolution of PEDV has been gathered [1,2]. However, the mechanisms underlying PEDV replication and pathogenesis have not yet been fully deciphered. Because laboratory procedures to isolate epidemic PEDV strains in cell culture and to manipulate the PEDV genome have proven challenging, a better understanding of such key viral processes has been hampered. In recent years, the latter task has been achieved by targeted RNA recombination technology [6] or reverse genetic approaches [7,8,9,10].

The reverse genetics (RG) system is a robust technological advancement in modern virology for studying virus molecular biology and pathogenic mechanisms, as well as for developing new and effective viral vectors and vaccines. To date, some RG methods have been developed for both LP-G1a and HP-G2b PEDV strains, in which infectious cDNA clones were constructed by in vitro ligation under the T7 promoter [7,8] or engineered into a bacterial artificial chromosome (BAC) plasmid under the cytomegalovirus (CMV) immediate-early promoter [9,10]. However, a versatile RG platform is not yet available for the Korean field strains of PEDV. In this study, we generated the first infectious cDNA clones of an HP-G2b Korean prototype strain, KNU-141112 [11], and its live attenuated vaccine (LAV) strain, S DEL5/ORF3 [12,13], in BAC under the control of a eukaryotic CMV promoter. Transfection of each BAC plasmid into Vero cells recovered progeny infectious PEDVs, icKNU-141112, and icS DEL5/ORF3, with phenotypic and genotypic characteristics comparable to each respective parental virus. In addition, using the DNA-launched HP-G2b PEDV infectious cDNA clone, pBAC-CMV-KNU-141112, as a backbone, two recombinant viruses were constructed that had either the ORF3 or S gene swapped with a reporter gene expressing the enhanced green fluorescent protein (EGFP) or a heterologous S gene expressing the variant S protein, respectively. Our BAC-based RG system will serve as a next-generation vaccine platform for PED, as well as for other emerging veterinary or human diseases.

## 2. Materials and Methods

### 2.1. Cells, Virus, and Antibodies

Vero cells (ATCC CCL-81) were cultured in alpha minimum essential medium (α-MEM; Invitrogen, Carlsbad, CA, USA) with 5% fetal bovine serum (FBS; Invitrogen) and Penicillin–Streptomycin (PS, 100×; Invitrogen). The cells were maintained at 37 °C in an atmosphere of humidified air containing 5% CO_2_ incubator. The Korean HP-G2b PEDV strain KNU-141112 and its LAV strain S DEL5/ORF3 were propagated in Vero cells with virus growth medium (α-MEM supplemented with 1% PS and 0.3% tryptose phosphate broth (TPB; Sigma-Aldrich, St. Louis, MO, USA), 0.02% yeast extract (Difco, Detroit, MI, USA), 10 mM HEPES (Invitrogen), and 5 µg/mL trypsin (USB, Cleveland, OH, USA) as described previously [13]. Each viral stock from the fifth passage in cell culture (KNU-141112-P5 and S DEL5/ORF3-P5) was prepared and served independently as a parental virus in this study. The PEDV N protein-specific monoclonal antibody (MAb) was obtained from ChoogAng Vaccine Laboratory (CAVAC; Daejeon, South Korea). The anti-ORF3 MAb was prepared in our laboratory using the recombinant ORF3 protein from PEDV strain KNU-141112 as the immunogen, which was produced using a mammalian cell-based transient gene expression system as described previously [14,15]. Antibodies specific for EGFP and β-actin were purchased from Abcam (Cambridge, UK) and Santa Cruz Biotechnology (Santa Cruz, CA, USA), respectively. The horseradish peroxidase (HRP)-conjugated goat anti-mouse IgG secondary antibody was obtained from Cell Signaling Technology (Danvers, MA, USA). Alexa Fluor 488-conjugated and Alexa Flour 594-conjugated goat anti-mouse secondary antibodies were acquired from Invitrogen.

### 2.2. Assembly of Full-Length PEDV cDNA Clones in BAC

The full-length nucleotide (nt) sequences of PEDV KNU-141112 and S DEL5/ORF3 (GenBank accession numbers KR873434 and KY825243, respectively) were independently used as a reference for the construction of each infectious cDNA clone in the BAC system using a three-step strategy as described previously [16]. In the first step, the whole genome of KNU-141112-P5 was divided into seven continuous fragments (1F to 7F) flanked by compatible restriction sites as follows: 1F, nt 1–2806; 2F, nt 2806–6808; 3F, nt 6808–13009; 4F, nt 13009–15880; 5F, nt 15880–23570; 6F, nt 23570–27185; 7F, nt 27185–28038 (Figure 1a). Since the large-sized fragment may be inherently unstable in the bacteria, the 3F and 5F were further subdivided into two (3F-1, nt 6808–9502; 3F-2, nt 9502–13009) or three (5F-1, nt 15880–19042; 5F-2, nt 19042–21540; 5F-3, nt 21540–23570) sub-fragments, respectively. The restriction sites, namely *Mlu*I, *Avr*II, *Pac*I, *Aat*II, *Cla*I, and *Bsp*T104I, which naturally occurred in the KNU-141112 genome, were selected, whereas two unique *Fse*I (15880) and *Asc*I (23570) sites in the fragment 5F were newly created at each terminus by introducing silent mutations (A15883C/A15884G and A23570G/T23573G/A23574C/A23576C, respectively), which were maintained as rescue markers. The locations of each enzyme site at the two ends of each fragment or sub-fragment are shown in Figure 1a. In addition, the CMV promoter sequence, with a *Bam*HI restriction site at its 5′ end, was added to the 5′ terminus of fragment 1F by overlapping PCR. The synthesized essential element sequences, including the 26-nt adenine sequence (26 pA), the hepatitis delta virus (HDV) ribozyme self-cleavage site (Rz), the bovine growth hormone (BGH) transcription terminal sequence, and a *Hind*III restriction site, were appended to the 3′ terminus of fragment 7F. Each of the fragments flanked by the respective restriction sites was PCR amplified using Platinum SuperFi Green PCR Master Mix (Invitrogen) and specific primer sets (available upon request), gel purified using the QIAEX II Gel Extraction Kit (Qiagen, Hilden, Germany), and then individually subcloned into the pCR-XL-2-TOPO vector (Invitrogen). All the subclones were verified by Sanger sequencing. Second, as a backbone to harbor the full-length PEDV clone, the plasmid pBeloBAC11 (NEB, Ipswich, MA, USA) was modified to insert a synthesized linker consisting of the unique restriction enzyme sites (*Mlu*I, *Pac*I, *Fse*I, *Asc*I, and *Bsp*T104I), between *Bam*HI and *Hind*III sites (Figure 1a), and the resulting BAC plasmid was named pBAC-CMV. Lastly, each subclone was digested from the corresponding pCR-XL-2-TOPO plasmid with specific enzymes collected with a gel extraction kit and sequentially assembled into the pBAC-CMV cassette vector step by step in an order (7F, 1F, 3F, 2F, 4F, 5F, 6F) using available restriction sites with DNA Ligation Kit LONG (TaKaRa, Otsu, Japan). The ligation mixture was electroporated into E. coli DH10B using a MicroPulser Electroporator (Bio-Rad Laboratories, Hercules, CA, USA). All the intermediate constructs were purified with NucleoBond Xtra Maxi Kit (Macherey-Nagel, Düren, Germany) and verified by restriction analysis and DNA sequencing before proceeding to the next assembly, and the resultant BAC plasmid was designated as pBAC-CMV-KNU-141112. An identical strategy was used to generate the full-length cDNA clone of S DEL5/ORF3 in the BAC system, named pBAC-CMV-S DEL5/ORF3 (Figure 1b). The genetic integrity of both PEDV infectious clones was verified by extensive restriction analysis and full-length nucleotide sequencing. DNA manipulation and cloning were performed according to standard procedures [17].

### 2.3. Rescue of Infectious PEDVs

The final BAC plasmids were prepared using the NucleoBond Xtra Maxi Kit (Macherey-Nagel, Düren, Germany). In addition, the pBud-PEDV N plasmid was constructed by cloning the PCR-amplified KNU-141112 N gene into the pBudCE4.1 expression vector (Invitrogen) and used to supplement the N gene in trans for increased virus recovery from the cDNA clone as described previously [7]. For in vitro transfection, Vero cells were seeded into a 6-well tissue culture plate at 3.5 × 10^5^ cells/well in Opti-MEM (Invitrogen) containing 10% FBS and then maintained in a humidified incubator at 37 °C and 5% CO_2_ for 24 h until approximately 90% confluency. The cells were co-transfected with 2 µg of the corresponding BAC plasmid (pBAC-CMV-KNU-141112 or pBAC-CMV-S DEL5/ORF3) and 1 µg of the pBud-PEDV N plasmid using 9 µl of FuGENE HD (Promega, Madison, WI, USA), according to the manufacturer’s protocols. At 18 h post-transfection, the cells were washed twice and then replenished with Opti-MEM supplemented with 5 µg/mL trypsin to assist in virus recovery and spread. The culture supernatants were harvested at 4–7 days post-transfection upon the visual observation of cytopathic effects (CPE) characterized by syncytium formation as described previously [11], and each infectious-clone-derived (ic) “passage 0 (P0)” PEDV was designated icKNU-141112-P0 or icS DEL5/ORF3-P0. The rescued P0 viruses were then subjected to plaque purification. A single, well-isolated plaque was picked, resuspended in virus growth medium, and stored as the P1 icPEDV stocks at −80 °C. Subsequently, each icPEDV-P1 was serially passaged on Vero cells for10 passages (P10), as described previously [13]. Each passage icPEDV was aliquoted and stored at −80 °C until use. The full-length genome of each P5 and P10 icPEDV stock was verified by Sanger sequencing, as previously described by our group [11,13,18,19].

### 2.4. Immunofluorescence Assay (IFA)

Vero cells grown on microscope coverslips placed in 6-well tissue culture plates were infected with parental (P5) and rescued (P10) PEDVs at a multiplicity of infection (MOI) of 0.1 for 1 h. At 24 h post-infection (hpi), infected cells were fixed with 4% paraformaldehyde for 10 min at room temperature (RT) and permeabilized with 0.2% Triton X-100 in PBS at RT for 10 min. The cells were blocked with 1% bovine serum albumin in PBS at RT for 30 min and then stained with PEDV N- or ORF3-specific MAb for 2 h. After washing five times in PBS, the cells were incubated at RT for 1 h with the Alexa Fluor 488-conjugated or 594-conjugated (only for icPEDV-EGFP) secondary antibody, followed by counterstaining with 4′,6-diamidino-2-phenylindole (DAPI; Sigma-Aldrich). The coverslips were mounted on glass microscope slides in a mounting buffer, and the stained cells were visualized under a Leica DM IL LED fluorescence microscope (Leica, Wetzlar, Germany).

### 2.5. Quantitative Real-Time RT-PCR (qRT-PCR)

For genomic RNA (gRNA) and subgenomic mRNA (sg mRNA) detection, Vero cells grown in 6-well tissue culture plates were infected with parental (P5) and rescued (P10) PEDVs at an MOI of 0.1. Total RNA was extracted from the lysates of infected cells at 24 hpi using the TRIzol Reagent (Invitrogen) and treated with DNase I (TaKaRa), according to the manufacturer’s instructions. The concentrations of the extracted RNA were measured using a NanoVue spectrophotometer (GE Healthcare, Piscataway, NJ, USA). The viral RNA was subjected to quantitative RT-PCR using a One Step PrimeScript RT-PCR Kit (TaKaRa) with primer/probe sets targeting gRNA and sg mRNA, as described previously [12,20,21]. The reaction was performed using a CronoSTAR 96 Real-Time System (Clontech, Mountain View, CA, USA) according to the manufacturer’s protocol. Copy numbers of each amplicon were determined by standard curves as described previously [12] and normalized with the mass of total cellular RNA. sg mRNA6 encoding N (sg mRNA-N) was also amplified in the sample as a control. The quantities of sg mRNAs were expressed as copies per 10^7^ copies of gRNA.

### 2.6. Identification of Rescue Markers in icKNU-141112 and icS DEL5/ORF3

Total RNA was extracted from the lysates of cells infected with parental (P5) and rescued (P10) PEDVs at 24 hpi using the TRIzol Reagent and treated with DNase I. RT-PCR was independently conducted to amplify 1800 bp (genomic positions 15277–17076) and 1000 bp (genomic positions 23008–24007 and 22993–23992 in KNU-141112 and S DEL5/ORF3, respectively) fragments spanning *Fse*I (15880) and *Asc*I (23570 and 23555), respectively, that had been created as molecular markers in the rescued icPEDV, but not in the parental virus. The presence of rescue markers in the PCR amplicons was determined by restriction enzyme analysis with *Fse*I and *Asc*I and nucleotide sequencing.

### 2.7. Virus Titration

Vero cells were infected with parental (P5) and rescued (P10) PEDVs in the presence of trypsin at an MOI of 0.1. The culture supernatant was collected at various time points (6, 12, 24, 36, and 48) and stored at −80 °C. Virus titers were measured by plaque assay using Vero cells and defined as plaque-forming units (PFU) per ml, as described previously [13]. In brief, Vero cells grown in 6-well plates were inoculated with 200 µL/well of 10-fold serially diluted virus suspensions containing trypsin and adsorbed for 1 h at 37 °C. The inoculated cells were overlaid with 2 mL of premixed virus growth medium and 1.5% Bacto Agar (Difco) and incubated at 37 °C for 2 days until appropriately-sized plaques were observed. At 48 hpi, the medium was removed, and the plaques were fixed with 7% paraformaldehyde and stained with 1% crystal violet in 5% ethanol.

### 2.8. Construction and Rescue of a Recombinant PEDV Expressing EGFP

The EGFP gene was inserted into the KNU-141112 genome by replacing ORF3, following the strategy described previously [9]. Briefly, the EGFP gene was PCR amplified with flanking PEDV sequences and then exchanged with the ORF3 coding region of fragment 6F by overlapping PCR to generate the ORF3 deletion (DEL) EGFP 6F construct. Due to the presence of the transcriptional regulatory sequence (TRS) of the E subgenomic mRNA (sg mRNA) in the 3′ terminus of ORF3, the 3′-terminal 22-nt sequences remained preserved in the fragment 6F to prevent interference with the expression of the E protein. The modified fragment 6F, named 6F-OFR3 DEL-EGFP, was subcloned into the pCR-XL-2-TOPO vector. The subclone 6F-ORF3 DEL-EGFP was digested with *Asc*I and *Bsp*T104I and gel purified. The pBAC-CMV-KNU-141112 plasmid was also digested with the same enzymes, and the original fragment 6F was replaced with the 6F-ORF3 DEL-EGFP harboring the EGFP gene via *Asc*I and *Bsp*T104I restriction sites. The final recombinant clone was verified by full-length nucleotide sequencing and designated pBAC-CMV-KNU-141112-EGFP. The rescue and subsequent cell culture passage of pBAC-CMV-KNU-141112-EGFP were performed as described above. The recombinant PEDV rescued from this BAC plasmid was named icPEDV-EGFP. The full-length genome of each P5 and P10 icPEDV-EGFP stock was verified by Sanger sequencing, and icPEDV-EGFP-P10 virus was subjected to fluorescence microscopy and virus titration, as described above.

### 2.9. Western Blot Analysis

Vero cells were grown in 6-well tissue culture plates for 1 day and were infected with icKNU-141112-P10 and icPEDV-EFGP harvested at the indicated passage history (P2–P10) at an MOI of 0.1. At 48 hpi, the cells were harvested in 80 µL of lysis buffer (0.5% Triton X-100, 60 mM β-glycerophosphate, 15 mM ρ-nitrophenyl phosphate, 25 mM MOPS, 15 mM MgCl_2_, 80 mM NaCl, 15 mM EGTA [pH 7.4], 1 mM sodium orthovanadate, 1 µg/mL E64, 2 µg/mL aprotinin, 1 µg/mL leupeptin, and 1 mM PMSF) and sonicated on ice five times each for 1 s. The homogenates were lysed on ice for 30 min and clarified by centrifugation at 15,800× *g* (Eppendorf centrifuge 5415R, Hamburg, Germany) for 30 min at 4 °C. In addition, cells were infected with icKNU-141112-P10 and icPEDV-EGFP-P10 and lysed at the indicated time points as described above. The total protein concentrations in the supernatants were estimated using a BCA protein assay (Pierce, Rockford, IL, USA). The cell lysates were mixed with 4× NuPAGE sample buffer (Invitrogen) and boiled at 70 °C for 10 min. Equal amounts of total protein were separated on a NuPAGE 4–12% Gradient Bis-Tris Gel (Invitrogen) under reducing conditions and electrotransferred onto an Immobilon-P membrane (Millipore, Bedford, MA, USA). The membranes were subsequently blocked with 3% powdered skim milk (BD Biosciences, San Jose, CA, USA) in TBS (10 mM Tris-HCl [pH 8.0], 150 mM NaCl) containing 0.05% Tween-20 (TBST) at 4 °C for 2 h and reacted with the primary antibody against EGFP, N, or β-actin at 4 °C overnight. The blots were then incubated with the HRP-labeled secondary antibody at a dilution of 1:5000 for 2 h at 4 °C. Finally, the proteins were visualized using enhanced chemiluminescence (ECL) reagents (GE Healthcare), according to the manufacturer’s instructions.

### 2.10. Detection of EGFP Fluorescence Intensity

Vero cells at 2 × 10^4^ cells/well were grown in 96-well tissue culture plates for 24 h and were mock infected or infected with icPEDV-EGFP-P10 at an MOI of 0.1. At the indicated time points, the cells were examined with a Leica DM IL LED fluorescence microscope, and pictures of the EGFP fluorescence images were obtained. Subsequently, the fluorescence intensity of EGFP was measured at 485/538 nm using a microplate fluorometer (Fluoroskan FL; Thermo Scientific, Waltham, MA, USA).

### 2.11. Clinical Sample Collection and Nucleotide Sequence Analysis

Clinical samples (feces and small intestine) submitted for laboratory diagnosis were collected from PED-affected swine farms located in eight different provinces nationwide from 2019 to 2022. The procedures for sample processing and RT-PCR for PEDV detection are described elsewhere [18,21,22,23].

The S glycoprotein gene sequences of PEDV-positive samples (*n* = 97) were determined using Sanger sequencing. Two overlapping cDNA fragments spanning the entire S gene of each isolate were amplified by RT-PCR as previously described [24]. The individual cDNA amplicons were gel-purified, cloned using the pGEM-T Easy Vector System (Promega), and sequenced in both directions using two commercial vector-specific T7 and SP6 primers and gene-specific primers. A total of 97 full S genes of PEDVs identified in this study have been deposited in the GenBank database under the accession numbers ON263410–ON263414, ON263422–ON263463, and OP186873–186932.

The sequences of fully sequenced S genes of global PEDV isolates were aligned using the ClustalX 2.0 program [25], and the percentages of amino acid sequence divergences were assessed using the same software. Phylogenetic trees were constructed from an alignment of the amino acid sequences using the neighbor-joining method and were subjected to bootstrap analysis with 1000 replicates to determine the percentage reliability values for each internal node of the tree [26]. A phylogenetic tree was generated using the Mega X software [27].

### 2.12. Construction and Rescue of a Recombinant PEDV Harboring a Heterologous S Gene

The variant S gene was introduced into the KNU-141112 genome by replacing the S gene following the strategy as described below. Since the S gene spans the two sub-fragments, 5F-2 and 5F-3, and the fragment 6F in the BAC plasmid, these three (sub)fragments were independently used as a template to replace the KNU-141112 S gene coding sequence with that of the heterologous GNU-2110 S gene, designated S51, as described below, using overlapping PCR. Each modified fragment, namely 5F-2-S51, 5F-3-S51, and 6F-S51, was subcloned into the pCR-XL-2-TOPO vector. To construct a recombinant KNU-141112 clone with the variant S51 gene, three sub-fragments, 5F-1, 5F-2-S51, and 5F-3-S51, were first digested from the corresponding pCR-XL-2-TOPO plasmid with their specific enzymes and ligated with the pBAC-CMV-KNU-141112 plasmid pre-digested with *Fse*I and *Asc*I. The fragment 6F-S51, harboring the remaining S51 gene and the KNU-141112 structural genes, was then swapped with the original 6F fragment in the above-modified BAC plasmid via *Asc*I and *Bsp*T104I restriction sites. The final recombinant plasmid exchanging the entire S gene sequence with that of GNU-2110 was named pBAC-CMV-KNU-141112-S51 and verified by full-length nucleotide sequencing. The rescue and subsequent cell culture passage of pBAC-CMV-KNU-141112-S51 were performed as described above. The nomenclature of a recombinant PEDV rescued from pBAC-CMV-KNU-141112-S51 was icPEDV-S51. The full-length genome of each P5 and P10 icPEDV-S51 stock was verified by Sanger sequencing, and icPEDV-S51-P10 virus was subjected to IFA and virus titration, as described above.

### 2.13. Serum Neutralization

The cross-neutralizing activity of antisera collected from sows vaccinated using second-generation G2b vaccines [12,21] was evaluated using a conventional virus neutralization test (VNT) in 96-well microtiter plates against icKNU-141112 and isPEDV-S51 as previously described [21], with minor modifications. In brief, Vero cells at 2 × 10^4^ cells/well were grown in 96-well tissue culture plates for 24 h. Each recombinant PEDV P10 stock was diluted in serum-free α-MEM to achieve 200 50% tissue culture infective doses (TCID_50_) in a 50 μL volume. Inactivated antiserum samples were serially diluted in α-MEM (1:4 to 1:512) and mixed with an equal volume of each virus at 37 °C for 1 h. One hundred microliters of the mixture were added to Vero cells and incubated at 37 °C for 1 h. After removing the mixture, the cells were rinsed with PBS five times and maintained in a virus growth medium at 37 °C in a 5% CO_2_ incubator for 2 days. The neutralizing endpoint titers were calculated as the reciprocal of the highest serum dilution that inhibited the virus-specific CPE by ≥80% relative to the controls in duplicate wells. The serum samples with neutralizing endpoint titers of ≥1:4 were considered to be positive for the PEDV-specific neutralizing antibody (NAb).

### 2.14. Structural Modeling and Sequence Alignment

The S glycoprotein 3-dimensional (3D) structure models of KNU-141112 and GNU-2110 were created based on the cryo-electron microscopy structure template with PDB accession code of 6vv5.1 using the SWISS-MODEL (https://swissmodel.expasy.org, accessed on 19 August 2022) online tool [28]. The S structure alignment of KNU-141112 and GNU-2110 was also performed by using the SWISS-MODEL server. In addition, the N-glycosylation sites were predicted by the NetNGlyc 1.0 server (http://www.cbs.dtu.dk/services/NetNGlyc/, accessed on 31 July 2022).

### 2.15. Statistical Analysis

All values are expressed as the mean ± standard deviation of the mean difference (SDM). Statistical analyses were conducted using the GraphPad Prism 7 software package (GraphPad Software, San Diego, CA, USA). *p*-values less than 0.05 were considered to be statistically significant.

## 3. Results

### 3.1. Construction of PEDV Infectious cDNA Clones in BAC

To develop two molecular clones encompassing the entire genome of HP-G2b PEDV and its LAV strain in BAC, we first synthesized seven contiguous viral fragments, designated 1F–7F, flanked with unique restriction sites commonly present or engineered at specific sites of the KNU-141112 and S DEL5/ORF3 genome (Figure 1) and subcloned each fragment into the pCR-XL-2-TOPO vector. The 5′ end of fragment 1F was fused to the CMV promoter so that the PEDV cDNA genome would be transcribed under the control of this promoter upon transfection. The 3′ terminus of fragment 7F was modified by fusion with the sequences containing 26 pA, HDV Rz, and BGH termination signals for the accurate processing of the PEDV genome 3′ end. Notably, synonymous substitutions were independently introduced at both termini of fragment 5F (genomic positions 15880 and 23570 (23555 for S DEL5/ORF3)) to generate two novel restriction sites, *Fse*I and *Asc*I, and were used as genetic markers to differentiate the recombinant from the parental virus. Next, the intermediate plasmid pBAC-CMV that contains a set of multiple enzyme restriction sites was generated as the backbone to facilitate the assembly of the full-length cDNA clone. Finally, each subclone was excised from the corresponding PCR cloning vector through appropriate restriction sites, and the full-length clone was assembled by sequential cloning of each DNA subclone into the multicloning site of the pBAC-CMV plasmid in the order of 7F, 1F, 3F, 2F, 4F, 5F, 6F. Using this three-step strategy, we successfully obtained two PEDV full-length genome clones in the BAC system, named pBAC-CMV-KNU-141112 (Figure 1a) and pBAC-CMV-S DEL5/ORF3 (Figure 1b).

### 3.2. Rescue of Infectious PEDVs from the cDNA Clone

To recover the RG-derived infectious virus, full-length BAC (pBAC-CMV-KNU-141112 or pBAC-CMV-S DEL5/ORF3) and pBud-PEDV N plasmids were transfected into Vero cells, and the rescued virus-induced CPE was monitored daily. Upon the appearance of the CPE of syncytial fusion, the culture supernatants were collected from each transfection and designated icKNU-141112 and icS DEL5/ORF3. These rescued P0 viruses were cloned by plaque purification and further propagated up to 10 passages to increase the virus titer. The P10 recombinant viruses were used to determine their phenotypic and genotypic properties. As shown in Figure 2a,b, icKNU-141112 and icS DEL5/ORF3 induced apparent CPE comparable to their respective parental viruses, such as cell fusion, syncytium formation, and detachment, in infected Vero cells (left panels), which were originally visible in transfected cells (P0). In contrast, the mock groups did not show any CPE, as verified by IFA. The replication of the recombinant viruses was confirmed by IFA using MAbs to PEDV N or ORF3 protein. Distinct N protein staining was distributed throughout the cytoplasm in the typical syncytial cells (Figure 2a, second panels). As we have shown previously, S DEL5/ORF3 causes vacuolation and syncytia in giant multinucleated cells that are more extensive and noticeable than in cells infected with KNU-141112 [13]. This phenomenon was reproduced in cells infected with the recombinant icS DEL5/ORF3 virus, as compared with the icKNU-141112-infected cells (Figure 2a,b; arrows). The S DEL5/ORF3 strain has also been reported to contain a large 46-nt DEL in an intergenic region (IGR) between S and ORF3 (S-ORF3 IGR; genomic positions 24775–24820), possibly resulting in interference with ORF3 expression [13]. As expected, ORF3 proteins were detected only in cells infected with the KNU-141112 virus and not in those infected with S DEL5/ORF3 and icS DEL5/ORF3 (Figure 2b, second panels). However, since the TRS of ORF3 is located upstream from the 46-nt DEL in the S-ORF3 IGR (Figure 3c), the sg mRNA3 encoding ORF3 (sg mRNA-ORF3) levels in the S DEL5/ORF3- and icS DEL5/ORF3-infected Vero cells remained unchanged compared to those in the KNU-141112- and icKNU-141112- infected cells (Figure 2e).

The identity of icPEDVs was further corroborated by restriction enzyme digestion and nucleotide sequencing of the fragments carrying genetic markers. In the genome of rescued viruses, six synonymous substitutions were introduced to create *Fse*I (15880) and *Asc*I (23570 or 23555) recognition sites. The 1800 bp and 1000 bp fragments covering each genetic marker were RT-PCR amplified from the parental and rescued virus strains. The respective fragments amplified from the parental KNU-141112 and S DEL5/ORF3 viruses could not be digested, whereas the amplicons from the rescued icKNU-141112 and icS DEL5/ORF3 viruses were digested by *Fse*I and *Asc*I into two fragments (1191 and 609 bp and 564 and 436 bp, respectively) (Figure 3a). These regions were also sequenced for genotype verification of the presence of marker mutations in the rescued viruses (Figure 3b). Further, the full-length genome sequences of two independent rescued viruses were analyzed to confirm their identities. The recombinant icKNU-141112 and icS DEL5/ORF3 viruses presented the same sequences as their respective cDNA clones. Moreover, the genotypic DEL signatures of the S DEL5/ORF3 strain, including the 15-nt DEL in the N-terminal domain (NTD) of S and the 46-nt DEL in the S-ORF3 IGR [13], were fully retained in icS DEL5/ORF3-infected cells with up to 10 serial passages (Figure 3c).

We next assessed the growth traits of the rescued viruses using a multistep growth assay with an MOI of 0.1. As seen in Figure 3d, icKNU-141112 and icS DEL5/ORF3 exhibited growth kinetics comparable to their respective parental viruses. However, icS DEL5/ORF3 replicated more efficiently than icKNU-141112, indicating that icS DEL5/ORF3 had a higher titer than icKNU-141112 at each time point, except for 6 hpi. Nevertheless, both icPEDVs reached peak titers over 10^6^ PFU/ml in cell culture. Furthermore, the plaque assay showed that the rescued viruses had plaque sizes and morphology similar to those of their respective parental viruses (Figure 3e). Interestingly, icS DEL5/ORF3 generally produced larger plaques than icKNU-141112, and this result is consistent with a previous study showing enlarged plaques of the parental S DEL5/ORF3 virus [13]. All these data indicate that two infectious full-length cDNA clones of wild-type HP-G2b and its LAV strains were successfully constructed in the BAC system, with each icPEDV and respective parental PEDV having the same phenotypic properties in vitro. 

### 3.3. Rescue and Characterization of Recombinant PEDV Expressing EGFP

The availability of the PEDV infectious cDNA clone established in this study opened the door to investigating its application as a viral vector to express a foreign protein. PEDV ORF3 is the sole accessory gene that is deleted or truncated from some field isolates and cell-adapted strains, suggesting its nonessential function in vitro or in vivo [6,13]. To test whether our DNA-launched rescue technique could be used to construct a fluorescently marked PEDV mutant, we introduced the EGFP gene into the ORF3 cassette using pBAC-CMV-KNU-141112, as depicted in Figure 4a. To this end, the coding sequence of ORF3 in the PEDV 6F fragment was replaced with that of EGFP, with the exception of the 3′-terminal 22 nucleotides containing the E TRS, in order to preserve E protein expression. After transfection of the EGFP-bearing recombinant clone (pBAC-CMV-KNU-141112-EGFP) and the PEDV N plasmid in Vero cells, CPE was observed (data not shown), and the rescued recombinant virus, named icPEDV-EGFP, was subjected to plaque purification and serial passages in Vero cells. To characterize the phenotype of icPEDV-EGFP, Vero cells were infected with icKNU-141112-P10 and mutant icPEDV-EGFP-P10. As shown in Figure 4b, fluorescence microscopy demonstrated that EGFP expression was observed only in cells infected with icPEDV-EGFP and not in those infected with icKNU-141112 (second panels). Notably, icPEDV-EGFP displayed distinct CPE, unlike icKNU-141112, showing extensive syncytium formation expressing EGFP (top panels, arrows), which was confirmed by N protein staining (third panels, arrows). Although both recombinant viruses showed comparable replication kinetics (Figure 4c), icPEDV-EGFP produced an enhanced plaque size compared to that of icKNU-141112 (Figure 4d). The identity of icPEDV-EGFP was further confirmed by restriction enzyme analysis and nucleotide sequencing (data not shown).

To assess EGFP gene stability and retention, the expression status of EGFP in cells infected with each passage of icPEDV-EGFP (P2–P10) was investigated by fluorescence microscopy and immunoblotting. As seen in Figure 5a, the strong expression of EGFP was stably retained through 10 serial passages in the Vero cells. Next, the EGFP expression kinetics of icPEDV-EGFP-P10 was explored at different time points post-infection by immunoblotting. Although the amount of viral N protein in icPEDV-EGFP-infected cells was comparable to that in cells infected with icKNU-141112, EGFP was detected only in icPEDV-EGFP-infected cells (Figure 5b). Similar to the expression kinetics of N, EGFP expression appeared at 12 hpi, gradually increased, and peaked at 48 hpi in icPEDV-EGFP-infected cells. The degree of gene expression intensity (fluorescence intensity) was further validated by fluorescence microscopy and fluorometry (Figure 5c). Collectively, the RG system could engineer the PEDV genome to generate a recombinant virus that stably expresses foreign proteins without altering viral protein expression phenotypes.

### 3.4. Genetic Monitoring of PEDV Epidemic Strains in South Korea

To further scrutinize whether the PEDV infectious cDNA clones could be manipulated to create recombinant PEDV that contains a heterologous S gene derived from a variant of interest (VOI), we first aimed to search for a candidate VOI that has a large number of S gene mutations. To this end, a molecular epidemiology study was conducted to analyze the complete S gene sequences of PEDV-positive samples collected nationwide through 2019–2022. Genetic analysis revealed that all the Korean isolates identified in this study were classified into HP-G2b, sharing a 96.3–99.4% amino acid (aa) identity with KNU-141112. The S sequences obtained were further compared with the reference sequences of the global PEDV strains, including all four genotypes (Figure 6a). In particular, one strain designated GNU-2110, isolated from Jeju Island in 2021, possessed the highest number (51) of mutations in the S gene, including a unique 3-nt (TTA) DEL at genomic positions 21025–21027, among the 2019–2022 epidemic strains, displaying approximately 4% variation (96.3% homology) and 1-aa shorter compared with KNU-141112. A total of 51 aa substitutions were randomly distributed in the S1 and S2 subunits, and among them, 16 changes were evident in NTD/S0 (residues 19–220) and “collagenase equivalent” (COE) (residues 502–641) neutralizing epitopes on S1 of HP-G2b PEDV [10,29]. Subsequent phylogenetic analysis indicated that GNU-2110 (red triangle) was the most distant from the HP-G2b Korean prototype strain KNU-141112 (black triangle), thereby forming a distinct branch within the subgroup G2b (Figure 6a). Thus, GNU-2110 was chosen as the VOI to introduce its S gene into the KNU-141112 genome using the RG approach.

### 3.5. Rescue and Characterization of Recombinant PEDV Expressing S Derived from VOI

To generate recombinant PEDV that encodes the S gene of the VOI, we swapped the indigenous S gene with the heterologous S derived from GNU-2110, named S51, using pBAC-CMV-KNU-141112 and constructed the S51-encoded recombinant clone (pBAC-CMV-KNU-141112-S51), as illustrated in Figure 6b. The recombinant virus, designated icPEDV-S51, was successfully recovered upon transfection and subsequently subjected to plaque purification and cell culture passages. We then investigated the virological properties of icPEDV-S51 in Vero cells. As shown in Figure 6c, both icKNU-141112 and icPEDV-S51 caused comparable CPE, including syncytium formation, in the infected Vero cells (top panels). The multiplication of the recombinant icPEDV-S51 virus was verified by IFA, showing well-defined N protein staining throughout the cytoplasm in syncytial cells akin to that observed in the icKNU-141112-infected cells (second panels). Furthermore, the growth kinetics and plaque morphology of icPEDV-S51 were comparable to those of icKNU-141112 (Figure 6d,e). The identity of icPEDV-S51 was further corroborated by restriction enzyme digestion and nucleotide sequencing (data not shown). These results indicated that the DNA-launched RG system could be used to construct recombinant PEDV that expresses heterologous S proteins originating from contemporary variants with growth ability identical to that of the parental virus.

We subsequently tested sow antisera previously elicited by second-generation G2b KNU-141112-based vaccines [12,21] to evaluate their neutralizing activity toward recombinant icKNU-141112 and icPEDV-S51 viruses using VNT (Figure 7). In accordance with previous studies, highly diluted antisera efficiently protected Vero cells infected with icKNU-141112 with a high NAb geometric mean titer (GMT) of 1:158, ranging from 64 to 512. Surprisingly, the antisera were less effective in inhibiting icPEDV-S51 infection with a NAb GMT of 1:22, which was almost 3-log_2_ lower than that against icKNU-141112. The NAb titers of each antiserum were further visualized with a heat map, which showed that the individual antisera exhibited significantly lower NAb titers against icPEDV-S51 compared to icKNU-141112. Collectively, our data revealed that the antisera limitedly cross-neutralized the recombinant PEDV harboring the heterologous S gene (S51), suggesting antigenic variations between KNU-141112 (2014 strain) and VOI (2021 strain).

The cross-neutralization data further implied that the considerable sequence dissimilarity found in NTD/S0 or/and COE domains on S1 between KNU-141112 and GNU-2110 could affect the virus-neutralizing activity. To visualize this hypothesis, we modeled the S glycoprotein 3D structures of KNU-141112 and GNU-2110. The structures of these S proteins were identically predicted as homotrimers with S1-NTDs exposed on the surface (Figure 8a). When aligned with the 3D structure of S of KNU-141112, the S51 (GNU-2110 S) displayed two major conformational changes on residues 55–74 and 130–147 residing on the NTD/S0 domain surface (Figure 8b). The prediction of N-linked glycosylation sites showed that the KNU-141112 S had 21 potential N-glycosylation sites defined by NxS/T (x is any residue excluding proline), whilst the GNU-2110 strain gained two additional putative sites at N130 and N1195 on the S1 and S2 regions, respectively (Figure 8c). Remarkably, the N130 glycan motif was predicted on the NTD/S0 domain of S51, which resulted from the occurrence of a 1-aa substitution (Ser-to-Asn (S130N)) and an isoleucine (Ile) residue DEL at positions 130 and 131, thereby altering the ‘SIKT’ residues to the N-glycosylation sequon ‘NKT’ at positions 130–132 (Figure 8c). Overall, we speculated that conformational and N-glycosylation site changes in S may influence the antigenicity of VOI (GNU-2110).

## 4. Discussion

The emergence or re-emergence of PEDV poses a significant global threat to animal health and long-lasting financial consequences for pork producers, which require cutting-edge research to improve our knowledge of virus biology and pathogenesis and to provide the basis for the development of intervention strategies. The advent of RG systems for most RNA viruses has aided viral genome manipulation and mutant construction for multiple purposes, ranging from the functional study of viral genes and protein to the development of vaccines. Due to the large size of the genome and the presence of the bacterial toxic domain, establishing such a state-of-the-art molecular tool for coronaviruses remains challenging. However, various alternatives to many traditional plasmid-based methods have been developed over the last decade [30]. Likewise, two different RG approaches have been reported for PEDV, and PEDV research has benefited from these availabilities [7,8,9,10,20]. The first approach is the T7 RNA polymerase-driven system, which comprises the in vitro ligation-based assembly of the full-length PEDV genome placed under the T7 promoter and in vitro transcription (IVT) and IVT RNA transfection [31]. Two research groups have recently used this technique to successfully produce infectious cDNA clones for Chinese and US HP-G2b strains [7,8]. However, this method has several inconveniences, including the costs and degradation risks involved in IVT and RNA manipulation [32]. The second approach is the DNA-launched BAC system that overcomes such drawbacks and is the most commonly used method, especially for viruses with large genomes [10]. Jengarn et al. and Li et al. independently managed to establish such a system for the Thai LP-G1a and Chinese LP-G1a and HP-G2b PEDV strains, respectively [9,10]. In this paper, we report the construction of two BAC-based DNA-launched molecular clones for a HP-G2b Korean strain of PEDV, KNU-141112, isolated in November 2014 [11], and its LAV strain, S DEL5/ORF3 [13]. The recovery of infectious PEDV was performed by transfecting each full-length cDNA clone directly into Vero cells, demonstrating that both recombinant cloned viruses, icKNU-141112 and icS DEL5/ORF3, were genetically stable and phenotypically comparable in vitro to their respective parental strains.

In previous studies, two PEDV isolates belonging to LP-G1a, designated AVCT12 (Thai strain) and CHM2013 (Chinese strain), were successfully recovered from their respective DNA-launched full-length cDNA clones in BAC [9,33]. Both AVCT12 and CHM2013 strains identically contain a 52-nt DEL in an IGR between S and ORFS (22-nt DEL at the end of S and 30-nt DEL at the start of ORF3), thereby encoding the C-terminal 7-aa truncated S gene and the impaired ORF3 [9,33]. Unexpectedly, RG failed to rescue the recombinant AVCT12 virus bearing full-length functional ORF3, suggesting an inhibitory role of ORF3 in PEDV replication in vitro [9]. In this study, however, the rescued icKNU-141112 efficiently produced an ORF3 protein comparable to its parental virus. Considering similar growth kinetics, the discrepancy between the two results is most likely related to the backbone sequence of the two AVCT12 and KNU-141112 strains. The Korean LAV strain, S DEL5/ORF3, which was obtained through serial passages and plaque cloning of HP-G2b KNU-141112 in cell culture, also possesses a 6-nt shorter DEL in S-ORF3 IGR than that in AVCT12 and CHM2013 strains, which removes the last 5-aa from the C-terminus of S and may interfere with ORF3 expression [13]. It is worth noting that the complete synthesis of the sg mRNA-ORF3 occurred since the TRS of ORF3 is unaffected by the DEL in S-ORF3 IGR. However, ORF3 protein production in cells infected with S DEL5/ORF3 and icS DEL5/ORF3 was ablated, indicating that the 46-nt DEL in S-ORF3 IGR truly interrupts the ORF3 gene reading frame. Although current knowledge on ORF3 function remains limited, ORF3 is proposed to be a multifuntional protein that acts as an ion channel and modulates host responses, which may contribute to virus replication and pathogenicity [34,35]. Notwithstanding the loss of ORF3, icS DEL5/ORF3 had a relatively faster replication kinetics than that of icKNU-141112, the same as shown in their respective parental viruses. The difference in growth phenotype is consistent with the observation that S DEL5/ORF3 and icS DEL5/ORF3 mostly made larger plaques than those of wild-type KNU-141112 and icKNU-141112. Collectively, the high growth of ORF3-null icS DEL5-ORF3 shown in this study confirmed that ORF3 is dispensable for PEDV propagation in vitro. 

Using the parental KNU-141112 infectious clone, we successfully generated the recombinant PEDV, icPEDV-EGFP, harboring EGFP at the ORF3 position that expresses EGFP instead of ORF3. The recombinant icPEDV-EGFP robustly produced EGFP and stably retained its intensity in infected cells for 10 serial passages while preserving a comparable growth ability relative to the parental icKNU-141112 virus. Consistent with previous studies [7,9], our results indicate that PEDV can tolerate the insertion of a foreign gene without negative selection. It is important to mention that despite comparable growth kinetics, ORF3-null icPEDV-EGFP produced syncytia and plaques greater than those from icKNU-141112. Our data from ORF3-null recombinant viruses (icS DEL5/ORF3 and icPEDV-EGFP) may dispute the nonessential role of ORF3 in cultured cells, as described above, suggestive of its relevance to PEDV cytopathology. Although we did not perform animal challenge experiments to assess pathogenic outcomes in vivo using icPEDV-EGFP, which can provide insights into understanding the correlation between ORF3 and PEDV virulence, accumulating evidence suggests that ORF3 may be an indispensable viral component to determine the pathogenic trait but insufficient to give rise to an attenuated phenotype of PEDV [7,12,13]. Nevertheless, the precise function of the PEDV ORF3 protein in vitro and in vivo remains to be further elucidated in future studies. More importantly, the current results demonstrate that the DNA-launched RG platform for BAC-based PEDV established in this study can make the construction of recombinant PEDV expressing foreign antigens possible, which is invaluable for comprehending fundamental viral processes as well as for pioneering vaccine or viral vector development. However, this platform could be hindered by potential hurdles regarding the genetic instability of the foreign gene since the loss of exogeneous inserts is highly frequent in positive-sense RNA viruses, including coronaviruses [9,36,37]. Although the molecular mechanism underlying exogeneous gene acceptance remains undetermined, several lines of evidence [6,9,36], including the present study, have demonstrated that the size (e.g., EGFP, mCherry, and luciferase) and location (e.g., ORF3) of the gene of interest introduced is a crucial factor for the genetic retention of recombinant coronaviruses. Thus, our ongoing task includes extensive investigations to replace ORF3 with various external non-coronaviral genes in the PEDV genome without transforming the stability and suppressing the replication of the recombinant viruses. In addition, the recombinant icPEDV-EGFP expressing EGFP can be useful in high-throughput screening in cell cultures for detecting NAb levels and evaluating therapeutic compounds.

PEDV isolation in cell cultures from clinical samples of infected pigs can be fastidious and labor extensive [11]. The virus isolation rate from PEDV-positive samples varies but is relatively very low, ranging from 2 to 10% [11,38]. In addition to difficulties in virus isolation, even isolated PEDV is often prone to lose infectivity upon further passages in cell culture [11,39]. The restricted success of the efficient isolation and propagation of PEDV in vitro is one of the key obstacles in performing varied PEDV research as well as developing effective vaccines in a timely manner. To overcome this basic constraint, the RG platform for BAC-based KNU-141112 was used to generate a recombinant icVOI by exchanging the S coding sequence of KNU-141112 with that of the field VOI (GNU-2110). We could then recover a VOI-like recombinant, icPEDV-S51, which encodes the full S gene from the field isolate GNU-2110 in the backbone of KNU-141112. GNU-2110 was selected as the candidate VOI because its S gene (named S51) was found to be the most genetically and phylogenetically divergent from that of KNU-141112, possibly leading to potential antigenic shifting. Remarkably, the antisera against KNU-141112 (NAb titers of ≥64) showed relatively weak reactivity and cross-neutralization toward heterologous S-expressing icPEDV-S51, suggesting that 51-aa variations in S51 are responsible for escaping from virus neutralization. Given that more than 30% of genetic drift in S51 was accumulated in NTD/S0 and COE neutralizing epitopes, some of these changes might contribute to the poor neutralization capacity of icPEDV-S51, albeit contributions from other mutations arising elsewhere in S cannot be excluded. Further structural analysis of PEDV S revealed that partial regions (residues 55–74 and 130–147) of the NTD/S0 neutralizing epitope on the S51 of GNU-2110 are conformationally heterologous to the counterpart in the S of KNU-141112. Compared to the KNU-141112 S protein, two other S glycan motifs were gained at sites N130 and N1195 on the GNU-2110 S protein (S51), and one “NKT” sequon at positions 130–132 was mapped within the conformation change on residues 130–147. More importantly, despite residing in a hypervariable region of the NTD/S0 domain, the “NKT” sequon (residues 130–132) is completely absent in the genome sequences of the other G2b strains identified, except for GNU-2110, but is often present in the LP-G1 strains. In addition, Li et al. (2017) described NAb escape mutants obtained by serial passaging of PEDV in vitro with high concentrations of S-specific neutralizing MAbs [10]. In this previous study, escape mutants resistant to each neutralizing MAb were found to display 3-aa substitutions in S at positions 100 (F100L), 129 (P129L), or 638 (V638G) that resided in NTD/S0 (residues 19–220) and COE (residues 502–641) domains on the S1 subunit [10]. Although none of these substitutions exist in S51, the “NKT” sequon (positions 130–132) is located next to a proline residue at position 129. Moreover, the glycan shield plays a critical role in the coronaviral S architecture, which is key to immune evasion and virus transmissibility [40,41]. Collectively, in addition to high mutations (almost 4% variation), the gaining of the glycan motif on the NTD/S0 neutralizing epitope domain may coat the S51 with the altered glycan shield topology and modify the S protein conformation, which in turn can afford some advantages (e.g., antigenic shift) that may allow the virus (i.e., GNU-2110) to evade host immune defenses, such as neutralizing antibodies, causing disease in immune animals.

In conclusion, this is the first report to describe the establishment of virulent and attenuated full-length cDNA clones of Korean HP-G2b PEDV in the BAC system, and it has been proven that DNA-launched molecular clones offer an effective recovery of infectious particles exhibiting phenotypic and genotypic traits analogous to parental viruses. Since the identification of virulence determinants or genes is key to understanding viral pathogenesis, the RG system for both HP and fully attenuated strains will aid future studies in investigating the importance of each genotypic DEL signature uncovered in S DEL5/ORF3 for the pathogenesis of PEDV. Given the continued emergence of new viral pathogens that endanger animal or/and human health, it seems obvious that the development of innovative, swift-responsive, and applicable intervention measures for disease control is necessitated. In this regard, our data documents that this RG platform can be used as a recombinant coronaviral vector vaccine that delivers a foreign antigen derived from any emerging veterinary or zoonotic viral pathogens in order to elicit protective immune responses. In South Korea, culturable Korean PEDV HP-G2b strains were unavailable until late 2014, one year after the HP-G2b virus was introduced into South Korea [11,42,43], and subsequently, new G2b-based killed vaccines and LAV were developed and marketed in 2017 and 2020, respectively. Considering this timeline, difficult and time-consuming virus isolation and attenuation will be the main hindrances to be surmounted for the rapid development of new vaccines against prospective variants emerging in the near future. Since the molecular clone was proven to be a rational backbone to embark the S gene of VOI, this platform will conquer such challenges associated with virus isolation and attenuation. This technical breakthrough furnishes a promising tool to enable the genetic engineering of any recombinant VOI with specific genetic markers that are predicted to affect an immune escape. This will serve as a template for the prompt generation of new live and killed vaccines and usher in a new era of PEDV vaccine research.

## Figures and Tables

**Figure 1 viruses-14-02319-f001:**
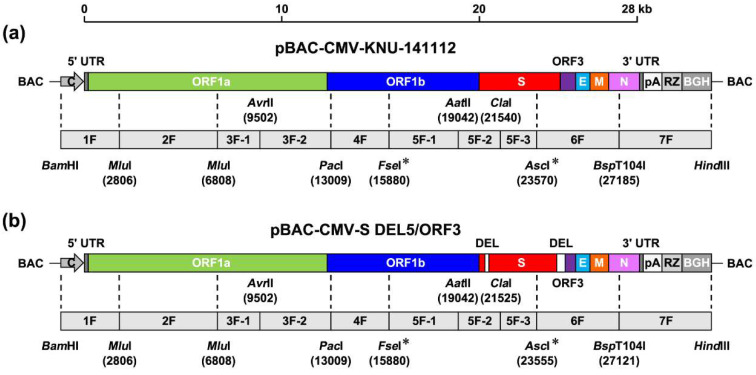
Strategy for the construction of two full-length molecular clones of PEDV strains, KNU-141112 and S DEL5/ORF3, in a BAC system. The illustration at the top of each panel represents the full-length cDNA genome organization of HP-G2b KNU-141112 (**a**) or LAV S DEL5/ORF3 (**b**). Viral genes (ORF1a, ORF1b, S, ORF3, E, M, and N) are illustrated by different colored boxes in each genome scheme. The CMV promoter (C), HDV ribozyme (Rz), and BGH transcription terminal sequences (BGH) are also shown in each genome diagram. 5′ and 3′ UTR, 5′ and 3′ untranslated regions; pA, poly(A) tail (26-nt). DEL on the S DEL5/ORF3 genome schematic represents the 15-nt (left) and 46-nt (right) DEL signatures. The second diagram in each panel represents seven overlapping cDNA fragments, termed 1F to 7F, covering the full-length genome of each strain, which were sequentially cloned into the plasmid pBAC-CMV to generate the PEDV infectious clones pBAC-CMV-KNU-141112 (**a**) and pBAC-CMV-S DEL5/ORF3 (**b**). The relevant restriction sites used for the assembly of the infection clone and their genomic positions (first nucleotide of the recognition sequence) are indicated. The *Bam*HI and *Hind*III sites existed in the multiple cloning site of the backbone vector pBAC-CMV. Two rescue markers, *Fse*I and *Asc*I sites, which were created by introducing silent mutations, are indicated by asterisks.

**Figure 2 viruses-14-02319-f002:**
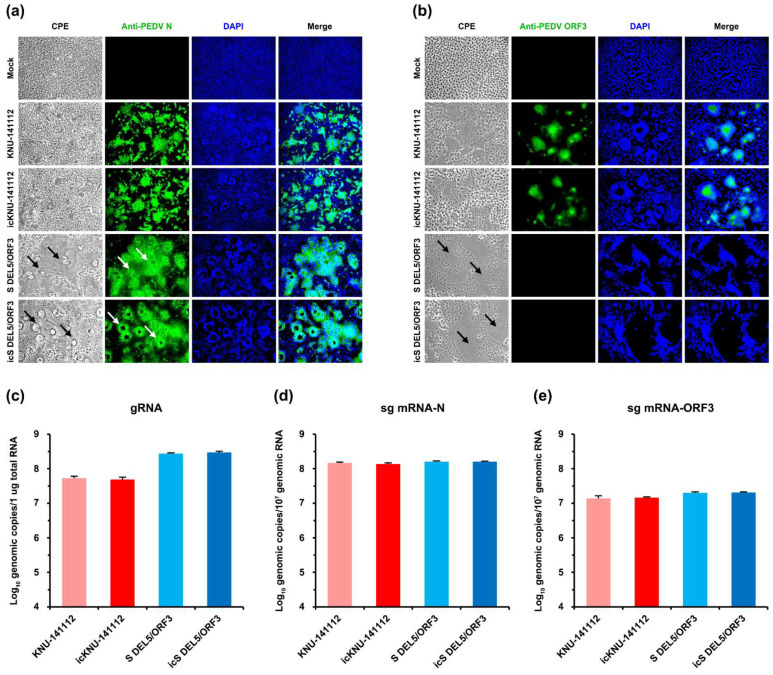
Identification of the virus rescued from each cDNA clone. (**a**,**b**) In vitro characterization of the infectious clone-derived viruses: icKNU-141112 and icS DEL5/ORF3. Vero cells were mock infected or infected with the parental and rescued viruses at an MOI of 0.1. PEDV-specific CPE was observed daily, and cells were photographed at 24 hpi using an inverted microscope at a magnification of 200× (left panels). For immunostaining, infected cells were fixed at 24 hpi and incubated with MAb against the PEDV N (**a**) or ORF3 (**b**) protein, followed by incubation with Alexa green-conjugated goat anti-mouse secondary antibody (second panels). The cells were then counterstained with DAPI (third panels) and examined under a fluorescence microscope at 200× (**a**) or 400× (**b**) magnification. Arrows indicate distinct syncytia and vacuolations in Vero cells infected with S DEL5/ORF3 or icS DEL5/ORF3. (**c**–**e**) Quantitative analysis of genomic RNA (**c**), sg mRNA-N (**d**), and sg mRNA-ORF3 (**e**) in PEDV-infected Vero cells at 24 hpi by qRT-PCR. The values shown are the mean of three independent experiments, and the error bars denote the SDM.

**Figure 3 viruses-14-02319-f003:**
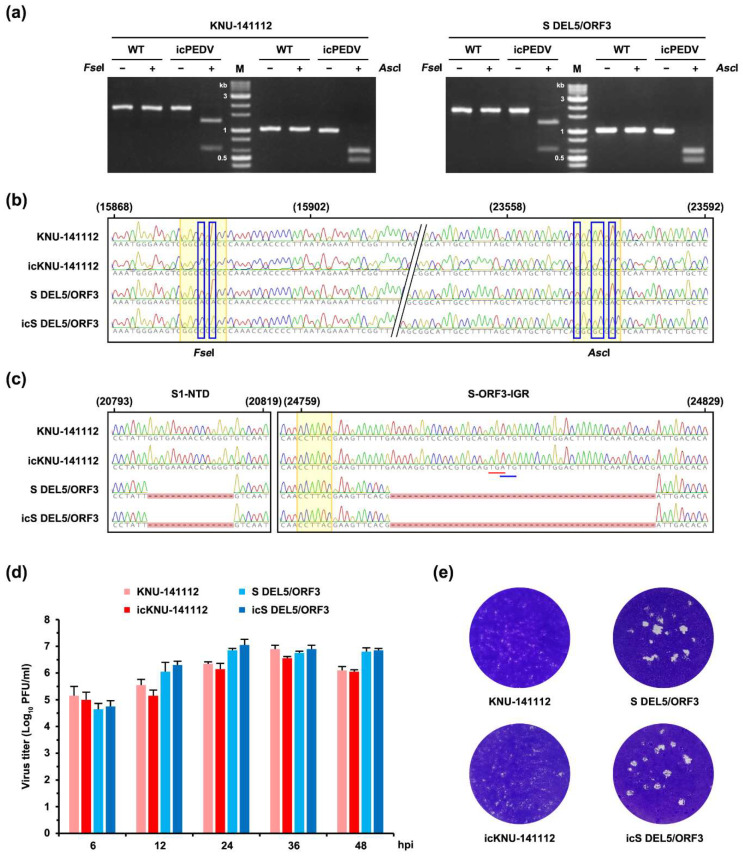
Genotypic and phenotypic characteristics of icKNU-141112 and icS DEL5/ORF3. (**a**) Confirmation of marker mutations in the rescued virus. Silent mutations were introduced to create two *Fse*I and *Asc*I sites at genomic positions 15880 and 23570/23555 (first nucleotide of the recognition sequence), respectively. Viral RNA was extracted from cells infected with each virus and subjected to RT-PCR analysis using primers covering each restriction site. Two 1800-nt and 1000-nt RT-PCR amplicons obtained from the parental (WT) and rescued (icPEDV) viruses of KNU-141112 (left panel) and S DEL5/ORF3 (right panel) were subsequently analyzed by *Fse*I and *Asc*I digestion and agarose gel electrophoresis. M, molecular mass marker (1-kb DNA ladder). (**b**) The sequence chromatograms of partial fragments (genomic positions 15868–15920 and 23542–23592) of the parental (KNU-141112 and S DEL5/ORF3) and rescued (icKNU-141112 and icS DEL5/ORF3) viruses. The *Fse*I and *Asc*I recognition sites are highlighted in yellow. The differences between the parental and rescued viruses are indicated by blue boxes. (**c**) Sequence analysis of the parental and rescued viruses at the genotypic DEL signatures in NTD of S and IGR between S and ORF3 of S DEL5/ORF3. The sequence chromatograms of partial fragments (genomic positions 20793–20819 and 24756–24829) of the parental and rescued viruses are shown. The dashes highlighted in red indicate the 15-nt DEL in S1-NTD (left panel) and the 46-nt DEL in S-ORF3 IGR (right panel). The stop codon in the S protein and the start codon in ORF3 are underlined in red and blue, respectively. The predicted 6-nt TRS (CCTTAC) of ORF3 is highlighted in yellow. (**d**) Growth kinetics of the parental and rescued viruses. At the indicated time points post-infection, culture supernatants were harvested, and virus titers were determined by plaque assay. The results are expressed as the mean of three independent experiments performed in duplicate, and the error bars show the SDM. (**e**) Representative image of plaque morphology of the parental (top) and rescued (bottom) viruses. Monolayers of Vero cells grown in 6-well plates were infected with each virus. The cells were overlaid with agarose and incubated for 2 days. Plaques were stained with crystal violet at 48 hpi and photographed.

**Figure 4 viruses-14-02319-f004:**
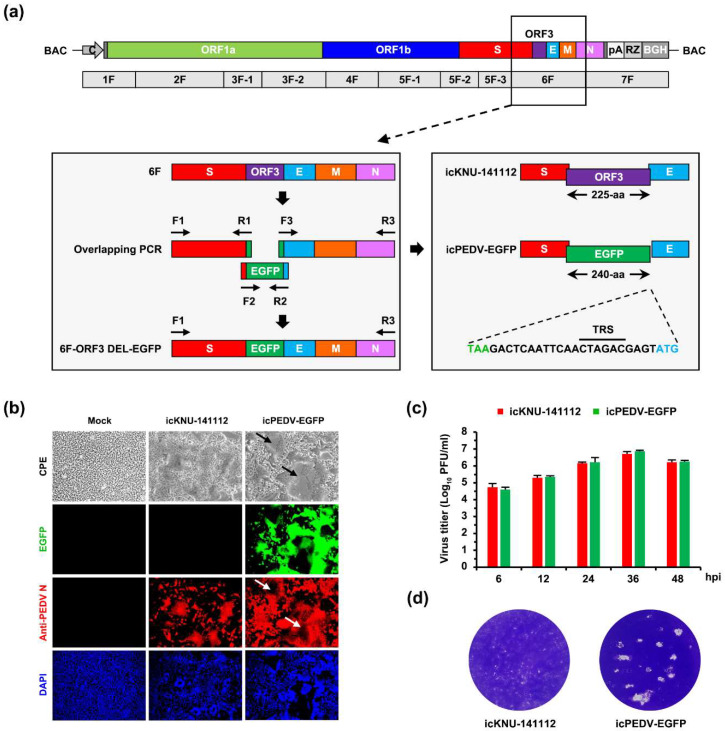
Construction and virological properties of recombinant icPEDV-EGFP. (**a**) Strategy for constructing the recombinant molecular clone KNU-141112-EGFP in BAC. The illustration on top represents the genome organization of KNU-141112, with different colored boxes symbolizing viral genes. The second diagram represents seven overlapping cDNA fragments (1F to 7F) comprising the KNU-141112 genome. Acronyms for viral genes and regulatory elements are described in Figure 1. The diagram on the left panel represents an enlarged version of fragment 6F, with different colored boxes indicating partial S (red), ORF3 (purple), E (cyan), M (orange), and partial N (pink). Three amplicons encompassing 6F, including EFGP (green) flanked by PEDV sequences, are illustrated with different colored boxes, which were individually PCR amplified using respective primer sets (hypothetically named F1-R1, F2-R2, and F3-R3), indicated by arrows and then used as templates for overlapping PCR using an “F1-R3” primer set to generate the 6F-ORF3 DEL-EGFP fragment depicted at the bottom. The modified fragment 6F was cloned into the plasmid pBAC-CMV-KNU-141112 to produce the recombinant infectious clone pBAC-CMV-KNU-141112-EGFP. In the right panel, the parental 225-aa ORF3 gene flanked by S and E sequences is depicted at the top. The second illustration represents the recombinant 240-aa EGFP flanked by S and E sequences. The 3′-terminal 22-nt sequence of ORF3, including the TRS of E, flanked by the stop codon in EGFP and the start codon in E, which were not removed to maintain E expression, is shown at the bottom. (**b**) Phenotypic characteristics of icPEDV-EGFP. Vero cells were mock infected or infected with icKNU-141112 and icPEDV-EGFP at an MOI of 0.1. PEDV-specific CPE was observed daily, and cells were photographed at 24 hpi using an inverted microscope at a magnification of 200× (top panels). For immunostaining, infected cells were fixed at 24 hpi and incubated with MAb against N protein, followed by incubation with Alexa red-conjugated goat anti-mouse secondary antibody (third panels). The cells were then counterstained with DAPI (fourth panels) and examined under a fluorescence microscope at 200× magnification. Arrows indicate distinct syncytia and vacuolations in Vero cells infected with icPEDV-EGFP. (**c**) Growth kinetics of icPEDV-EFGP. At the indicated time points post-infection, culture supernatants were harvested, and virus titers were determined by plaque assay. The results are expressed as the mean of three independent experiments performed in duplicate, and the error bars show the SDM. (**d**) Plaque morphology of icPEDV-EGFP. Representative plaques of Vero cells infected with icKNU-141112 (left) and icPEDV-EGFP (right) at 48 hpi are shown.

**Figure 5 viruses-14-02319-f005:**
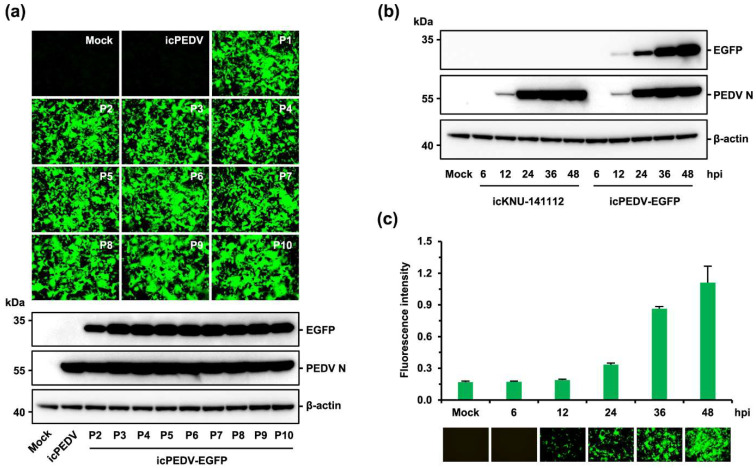
Genetic stability and retention of the EGFP gene of icPEDV-EGFP. (**a**) EGFP expression of icPEDV-EGFP during serial passages. Vero cells were mock-infected or infected with icKNU-141112 (icPEDV) and 100-fold diluted icPEDV-EGFP harvested at the indicated passage history (P1–P10) and observed at 48 hpi with a fluorescence microscope at a magnification of 100× (upper panels). Whole-cell lysates were prepared promptly after fluorescence microscopy and subjected to Western blotting with an antibody specific for EGFP or PEDV N. The blot was also reacted with mouse MAb against β-actin to confirm equal protein loading (bottom panel). (**b**) EGFP expression kinetics of icPEDV-EGFP. Vero cells were mock infected or infected with icKNU-141112 and icPEDV-EGFP at an MOI of 0.1. Cell lysates were prepared at the indicated time points and subjected to immunoblotting of an antibody against EGFP (top panel), PEDV N (second panel), or β-actin (bottom panel). (**c**) Fluorescence intensity kinetics of icPEDV-EGFP. Vero cells were mock infected or infected with icPEDV-EGFP at an MOI of 0.1. At the indicated time points, the EGFP fluorescence emission was detected by fluorometry, and the kinetics data were plotted. Green fluorescence images acquired with a fluorescence microscope at a magnification of 100× are presented at the bottom. The values shown are the mean of three independent experiments, and the error bars denote the SDM.

**Figure 6 viruses-14-02319-f006:**
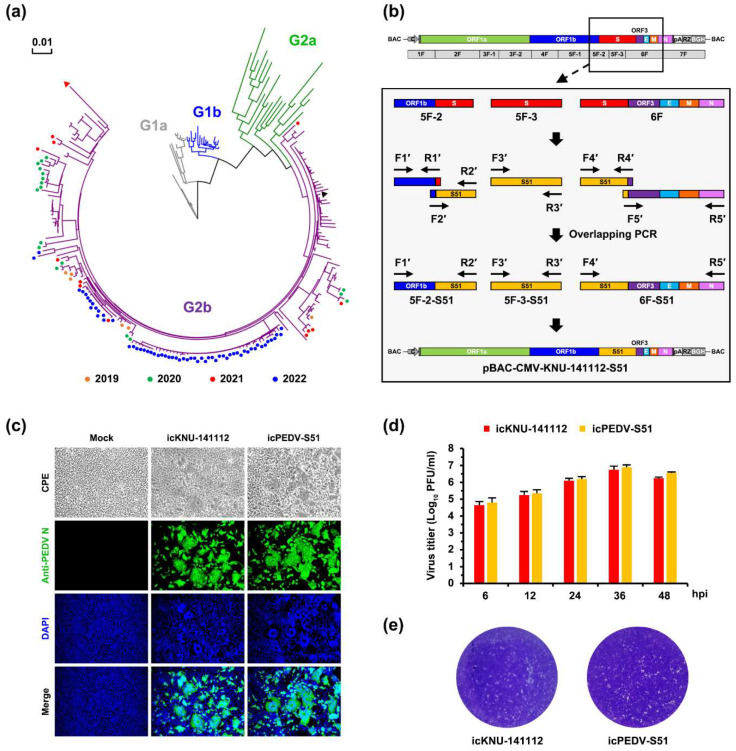
Construction and virological properties of recombinant icPEDV-S51. (**a**) Phylogenetic analysis based on the full S genes of the PEDV strains. Different colored dots indicate the PEDV strains chronologically identified in this study in 2019 (orange), 2020 (green), 2021 (red), and 2022 (blue). A red triangle indicates the VOI (GNU-2110) identified in this study, whereas a black triangle indicates the HP-G2b Korean prototype strain (KNU-141112). Four genotypes, G1a (gray), G1b (blue), G2a (green), and G2b (purple), are indicated. A scale bar indicates the number of nucleotide substitutions per site. (**b**) Strategy for constructing the recombinant molecular clone KNU-141112-S51 in BAC. The illustrations on top represent the genome organization of KNU-141112, which was divided into seven overlapping cDNA fragments (1F to 7F). Acronyms for viral genes and regulatory elements are described in Figure 1. The second diagram represents an enlarged version of the sub-fragments 5F-2 and 5F-3 and fragment 6F, spanning the full S gene, with different colored boxes indicating partial ORF1b (blue), S (red), ORF3 (purple), E (cyan), M (orange), and partial N (pink). Five amplicons covering 5F-2, 5F-3, and 6F, including S51 (yellow), are illustrated with different colored boxes, which were individually PCR amplified using respective primer sets (hypothetically named F1′-R1′ through F5′-R5′) indicated by arrows and then used as templates for overlapping PCR using primer sets (e.g., ‘F1′-R2′’ and ‘F4′-R5′’) to generate the 5F-2-S51, 5F-3-S51, and 6F-S51 (sub)fragments, respectively. These modified (sub)fragments were sequentially cloned into the plasmid pBAC-CMV-KNU-141112 to produce the recombinant infectious clone pBAC-CMV-KNU-141112-S51, as depicted at the bottom. (**c**) Phenotypic characteristics of icPEDV-S51. Vero cells were mock infected or infected with icKNU-141112 and icPEDV-S51 at an MOI of 0.1. PEDV-specific CPE was observed daily, and cells were photographed at 24 hpi using an inverted microscope at a magnification of 200× (top panels). For immunostaining, infected cells were fixed at 24 hpi and incubated with MAb against N protein, followed by incubation with Alexa green-conjugated goat anti-mouse secondary antibody (second panels). The cells were then counterstained with DAPI (third panels) and examined under a fluorescence microscope at 200× magnification. (**d**) Growth kinetics of icPEDV-S51. At the indicated time points post-infection, culture supernatants were harvested, and virus titers were determined by plaque assay. The results are expressed as the mean of three independent experiments performed in duplicate, and error bars show the SDM. (**e**) Plaque morphology of icPEDV-S51. Representative plaques of Vero cells infected with icKNU-141112 (left) and icPEDV-S51 (right) at 48 hpi are shown.

**Figure 7 viruses-14-02319-f007:**
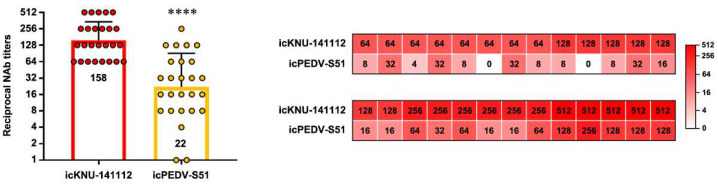
Cross-neutralization between icKNU-141112 and icPEDV-S51. Antiserum samples were subjected to VNT using icKNU-141112 and icPEDV-S51. NAb titers against icKNU-141112 and icPEDV-S51 are graphed as geometric mean titers (GMT) with geometric standard errors. The numbers within the bars show the GMT in the group. Values are representative of the mean from three independent experiments performed in duplicate, and the error bars denote the SDM. **** *p* < 0.0001. Individual NAb titers in each antiserum against icKNU-141112 and icPEDV-S51 are presented with a heat map on the right. The rainbow color from white to red corresponds to the NAb titer from low to high, respectively. The numbers within the map indicate the reciprocal NAb titers.

**Figure 8 viruses-14-02319-f008:**
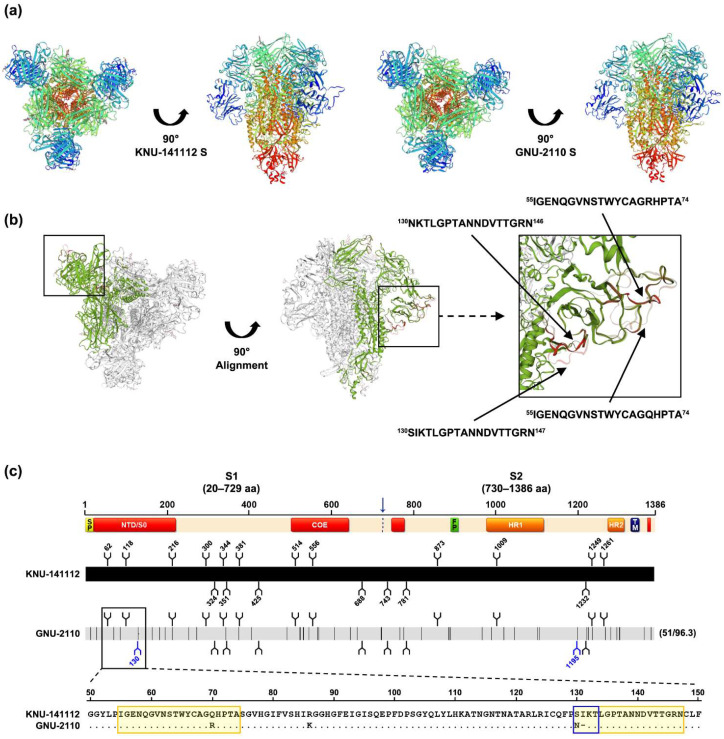
3D structural modeling and sequence alignment of KNU-141112 S and GNU-2110 S51. (**a**) 3D models of the S structure of KNU-141112 and GNU-2110. The predicted S glycoprotein 3D structural models were generated and observed at the top (left) and front (right) sites. (**b**) 3D structural alignment of S between KNU-141112 and GNU-2110. The 3D structures of the two S proteins were aligned and observed on the top (left) and front (right) sites. The region showing structural variations in the S protein between KNU-141112 and GNU-2110 is boxed, and its enlarged version is presented on the right. The residues with structural variations are shown in gray for KNU-141112 and magenta for GNU-2110. (**c**) Schematic diagram of genetic variations in the S gene between KNU-141112 and GNU-2110, showing the predicted positions of N-linked glycosylation. The top illustration represents the organization of the S protein, featuring S1 and S2 subunits that contain a signal peptide (SP), an N-terminal hypervariable domain (NTD), a fusion peptide (FP), heptad repeat regions (HR1 and HR2), and a transmembrane domain (TM). Red-highlighted areas in the diagram depicting the S protein represent four neutralizing epitopes (NTD/S0, residues 19–220; COE, residues 502–641; residues 744–774; residues 1371–1377) of HP-G2b PEDV. Potential N-glycosylation amino acid sites (NxS/T, where x ≠ P) are shown as branches. Lightly shaded areas are identical to KNU-141112, and the vertical black bars represent one amino acid sequence that is divergent from that of KNU-141112. The thin horizontal dashed line indicates a deleted amino acid residue. The digits in parentheses on the right indicate the number of amino acid changes and the percent identity compared with KNU-141112. Two additional glycosylation sequons at N130 and N1195 on S51 of GNU-2110 are indicated by blue branches. The amino acid sequence alignment of the partial S1-NTD domain (residues 50–150) between KNU-141112 and GNU-2110 is presented at the bottom. The residues with structural variations described in Figure 8b are highlighted in yellow, whereas the unique N-glycan motif (NKT) gained at positions 130–132 of GNU-2110 is marked with a blue box.

## Data Availability

The data that support the findings of this study are available from the corresponding author upon reasonable request.

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
