# Peer review of "Development of a Next-Generation Vaccine Platform for Porcine Epidemic Diarrhea Virus Using a Reverse Genetics System"

_viruses, 2022, doi:10.3390/v14112319_

Round 1

Reviewer 1 Report

The manuscript by Jang et al. reports construction of BAC-based PEDV infectious clones for Korean strains and shows that they can be used as important tools to study viruses. While this work is scientifically sound and meticulous and very well-written, the originality/novelty and contribution to the field of this work is arguably falling short. Reverse genetics of PEDV has been established for a while, and now is routinely used as part of the experiments to characterize viral genes/mutation/host factor’s effects toward viruses, not as the main message of the paper. The method by which they constructed the infectious clone was also not novel. The only really novel and interesting results of this paper is in Figure 7 when the authors used rg-derived recombinant PEDV to study cross-neutralization by antisera from vaccinated pigs in Korea. Therefore, I am not convinced that this manuscript would fit with the issue “State-of-the-Art Animal Virus Research in South Korea”. If the author expanded on the recombinant S51 as a vaccine candidate or characterized it further, maybe something more interesting and novel would be revealed.

Specific comments:

  1. Insertion of a fluorescent marker in PEDV infectious clones particularly at the ORF3 site in BAC has been done before but there is no citation. Line 476.

  2. Lines 737-738, PEDV ORF3 research has come a long way after the 2012 reference (ref 34) the authors cited. Maybe they would like to update and rewrite this part a little bit?

  3. Lines 754-755, this is a bit unclear. Lack of ORF3 expression is correlated with extensive and large syncytia. I understand that this is a clearly different phenotype from when the virus carries and expresses ORF3, but using language like “...suggestive of its (ORF3’s) essential role in PEDV cytopathology” seems counterintuitive when the viruses lacking ORF3 show more extensive CPE.

  4. Lines 794-818. Although it is quite enticing, as the authors also took this path in the paper, to attribute the difference in neutralizing activity by antisera between KNU-141112 and S51 viruses to those changes in NTD/CoE areas of Spike, this is far from certain. Conversely, almost 70% of the changes occur elsewhere in spike and might lead to decrease in neutralizing activity in the antisera. Perhaps a recognition of this possibility can give readers a more balanced view.   

  5. Figure 6, it would be very nice to see a blot for spike expression during viral infections of KNU-141112 and S51.

Reviewer 2 Report

In this paper, the authors constructed a series of PEDV mutant strains, including attenuated strain with ORF3 deletion, insertion strain with EGFP substitution of ORF3, and recombinant strain with S substitution. The rescued strains were also assayed for biological properties and analyzed for cross-neutralizing antibody potency differences. Overall, the article is informative and has some application prospects.

1. The Figure 3d shows that the virus titer reaches about 105 at 6hpi, which is unreasonable. For example, in Figure 5b, the viral protein was barely detectable at 6hpi, indicating that the virus titer was very low at this time. The same data is also questionable, for example, in Figure 4c and Figure 6d.

2. Figure 4d, icKNU has a significant difference in plaque size compared to icEGFP, while the growth kinetics are similar. Please explain the reason. Because the similar phenomenon is also present on S DEL5/ORF3, but its replication dynamics level is indeed higher than that of KNU.

Reviewer 3 Report

Authors reported a next-generation vaccine platform for PEDV by using a reverse genetics system. They used two strains KNU-141112 and DEL5/ORF3 to prepare reconstituted viruses. And they concluded that the established molecular clones provide key infrastructural frameworks for developing new vaccines and coronaviral vectors.

1. Remove EGFP in keywords.

2. Check typing mistakes, e.g. CO2 in Section 2.1.

3. Are enlargement factors in Fig 2 a and b the same? It is necessary to show IF pictures with scaleplates.

4. Show the molecular weight of the marker in Fig 3 a.

5. In all western blotting results, it is necessary to show the molecular weight of the detecting protein.

Round 2

Reviewer 1 Report

Thank you for the revision.

Reviewer 3 Report

The revision is acceptable.